# Physicochemical Evaluations of Diets, Rumen Fluid, Blood and Faeces of Beef Cattle under Two Different Feedlot Systems

**DOI:** 10.3390/ani12223114

**Published:** 2022-11-11

**Authors:** Pedro Malafaia, Vinícius Carneiro de Souza, Diogo Fleury Azevedo Costa

**Affiliations:** 1Department of Animal Nutrition and Pastures, Federal Rural University of Rio de Janeiro (UFRRJ), Seropédica 23897-000, RJ, Brazil; 2Department of Animal Science, University of California, Davis, CA 95616, USA; 3Institute of Future Farming Systems, Central Queensland University, Rockhampton, QLD 4701, Australia

**Keywords:** ancillary tests, digestive disorders, feedlot practices, ruminal acidosis

## Abstract

**Simple Summary:**

Mitigating the toll of nutritional disorders in feedlot systems can become difficult when the issue is not fully understood or correctly monitored. The physicochemical characteristics of diets and faeces were strongly correlated with digestive disorders in two distinct beef cattle feedlot systems. Ruminal and faecal pH, and particles size distribution in the faeces used in combination with information of the diet nutritional composition and fibrosity worked as good indicators for presence of digestive disorders. The use of these ancillary tests is proposed here as tools to facilitate best practice management protocols aiming to reduce the risks of subacute rumen acidosis.

**Abstract:**

The physicochemical characteristics of diets and faeces were evaluated in combination with data of rumen fluid and blood lactate collected from two distinct feedlot systems in Brazil to understand the causes and correlations to digestive disorders in these production systems. The data were collected during two visits to a finishing system which fed about 80,000 head per year, and four visits to two properties that fed 150 to 180 straight bred Nellore bulls per year to be sold as stud cattle. The findings suggest that ruminal acidosis occurred when there was high intake of starch-rich concentrate, and that subacute rumen acidosis (SARA) most likely occurred in situations where more than 4% of faecal dry matter was excreted as particles larger than 4 mm. The latter were associated with diets having less than 15% of particles smaller than 8 mm and faecal pH under 6.30. It is concluded that ancillary tests, such as ruminal and faecal pH, and particle size distribution in the faeces, can potentially be used in combination with information on diet nutritional composition and a series of best practice management protocols to increase not only animal productivity but to reduce the risks of SARA and ensure the welfare of animals.

## 1. Introduction

Raising cattle in feedlots is an important strategy for beef production in Brazil and in many other parts of the world. Brazilian feedlots systems are adopted for one of two purposes: (1) rapid animal growth/fattening before slaughter, or (2) to allowing faster growth of purebred males to be used as sires for cow-calf systems. The more traditional system is referred as a finishing feedlot, whilst the second one is known as for its use in the preparation of stud bulls.

The correct identification of nutritional disorders is not an easy procedure, and most Brazilian feedlots suffer from lack of adequate support on this matter. The real dimension of the issue is not always correctly measured because of a high prevalence of subclinical or subacute forms of these digestive disorders, and because of the large number of animals to be observed/examined by veterinarians on a daily basis. These digestive disorders cause great economic losses in beef cattle finishing systems, especially where high-grain diets are used [1,2]. 

Information on the particle sizes of diets and faeces have the potential to be used to make inferences on whether ruminal fermentation is adequate, and to explain some behavioral disorders in ruminants [1,3]. It is well known that rumination and saliva production are interconnected physiological events that contribute to adequate rumen dynamics, health, and microbial growth on the substrates, mainly in terms of fibrous compounds [4,5]. Research has indicated that feed particles inside the bovine rumen-reticulum need to be reduced to a size between 1.5 to 2.0 mm to pass through the reticulo-omasal orifice and reach the abomasum, from where they escape to the faeces [6]. However, most of the studies available on particle dynamics have been conducted with animals without health problems that have been adequately fed [6,7]. Nevertheless, when cattle have subacute rumen acidosis (SARA), or are under strong heat stress, a greater proportion of feed particles larger than 4 mm often appear in their faeces [1,3]. Diets rich in concentrates and poor in physically effective fiber (peNDF) may result in faeces with larger particles, due to the reduction in rumination time and potential increase of rumen fractional outflow rate. This implies less saliva production and, consequently, decreased ruminal pH and impairments in the multiplication and activity of the rumen microbiota, especially fibrolytic bacteria [8]. The main fibrolytic bacteria are intolerant to low ruminal pH, and rumen fiber digestion drastically decreases when pH drops below critical values [9]. In these situations, the reduction in the time spent ruminating, and the increase in the rate of passage of digesta through the rumen, cause the escape of larger particles to the faeces.

Our hypothesis is that fecal particle size distribution, associated with ruminal and faecal pH, can be an indicator of the rumen health of feedlot cattle in the finishing phase. If the hypothesis is confirmed, these could be useful ancillary tests in the diagnosis of nutritional disorders (e.g., SARA), especially in the subclinical phase when the identification of symptoms is difficult, as they only exhibit unspecific and/or discrete signs. Given the above, this study aimed to evaluate the physicochemical parameters of diets and faeces, rumen fluid, and blood lactate in beef cattle raised under two different types of feedlot systems in Brazil. The authors hope this can be used to aid in the identification of SARA, the most prevalent digestive disorder in feedlot cattle.

## 2. Materials and Methods

### 2.1. Finishing Feedlot (Traditional System)

Data were collected in October 2012 and May 2013 from a feedlot located in the state of Goiás, Brazil, which bought in and finished about 80,000 head per year. This feedlot finished straight and crossbred Nellore cattle. The diets were composed of grains and by-products and mombaça grass silage (M. maximum cv. Mombaça) as roughage (Table 1). The feeding period varied from 90 to 110 days. Two adaptation diets (A1 and A2) with decreasing amounts of roughage were fed for one week each. From the end of the second week (A2 phase) until their slaughter, animals were fed the finishing diet (F) (Table 1). 

The final finishing diets contained about 80–85% concentrate (on DM basis) and were fed daily at 8:00, 10:00, 12:00, 14:00, 16:00 and 20:00 h. 

During each visit, samples of the diets A1 and A2 (day 3 of each adaptation phase) and of the finishing diet (days 30 and 60) were collected. As the total diet was fed several times throughout the day, subsamples of A1, A2 and F were collected directly from the bunks at the moment of feed delivery. These subsamples were bulked at the end of the day, identified and frozen until further analysis. 

The particle size evaluations of the total diets were performed with small adjustments in the methodology described by Pereira et al. [10]. Samples were thawed, and 100 g was placed in an electric particle separator, mounted with 8.0, 4.0 and 2.0 mm sieves, as well as a bottom pan compartment. The device was turned on and left working for 10 min at 5000 oscillations/minute (80% of vibration intensity). After this procedure, the sieves were placed in an oven (60 °C until constant weight) and the weight of the sample retained on each sieve was expressed as a percentage of the original dry matter (DM) of the sample subjected to particle size separation. The remainder of each sample was oven dried (60 °C until constant weight) and then ground in a knife mill to pass through a 1 mm screen sieve. Ground samples were analyzed for DM, crude protein (CP), ether extract (EE), neutral detergent fiber (NDF), acid detergent fiber (ADF), lignin, crude ash, calcium (Ca) and phosphorus (P), according to the protocols described by Silva and Queiroz [11]. 

The estimates of non-fibrous carbohydrates (NFC) were made according to the following equation: NFC = OM − (PB + EE + NDFap), where NDFap constitutes the insoluble fraction in neutral detergent fiber corrected for ash and protein, and OM is equivalent to organic matter. The chemical composition of the diets is presented in Table 2.

In October 2012 and in May 2013, ten samples of faeces (±80 g) were randomly collected from the floor of the five pens from A1, A2 and F groups. The latter samples were checked to assure they had a similar faecal score and were collected from cattle with similar liveweight (LW) that had defecated at the moment of collections. The upper portion of each fecal sample was collected leaving the part touching the soil behind to avoid sample contamination. These samples were identified, stored in plastic bags and frozen for further analyzes. After thawing, fecal samples were analyzed for pH using a portable pH meter. Nine grams of each fecal sample were placed in a beaker containing 60 mL of distilled water and shaken using a glass stick at 0, 15 and 25 min. At 30 min, the pH meter electrode was inserted in the solution and the pH measured [11].

The particle size separation of the faeces was carried out in a wet sieving particle separator using sieve mesh sizes 1.18, 2.0 and 4.0 mm, respectively. Prior to this process, the faeces were thawed for 12 h at room temperature and a 100 g sample incubated for 1 h in 500 mL of distilled water containing 50 mL of neutral detergent solution. Every 15 min the sample was gently shaken with a glass stick. This previous incubation in water with neutral detergent aimed to dissolve the fecal mucus and reduce the agglutination of fecal particles. The wet sieving apparatus was set to vibrate at 4000 oscillations/minute for 10 min and under a water flow of 1.0 to 1.5 L/min. The sieves containing the fecal particles were oven dried (55 °C for 24 h) and then weighed. The dry faeces, retained in each mesh size, were described in percent of the total fecal dry matter sieved.

### 2.2. Feedlots for Production of Stud Bulls 

Data were collected in two farms located in Valparaiso (NC Farm, May and July 2013) and Guararapes (BA Farm, April and July 2013), in the State of São Paulo, Brazil. Both farms raise purebred Nellore cattle and sell approximately 150 to 180 stud bulls per year to be used as sires. The selection process was based on calf pedigree and their weight at 120 and 210 days (at weaning). After weaning, the animals were raised on pasture until reaching 480 to 500 kg LW. Afterwards, they were fed in pens for 120 to 150 days to reach target weights of 650 to 680 kg LW before sale. In these systems, groups containing 30 to 40 animals were allocated into open-dirt pens (40–45 m^2^/head) with a non-covered bunk line and a water trough. The pens were scraped every 15 to 20 days to remove accumulated faeces.

Feeding occurred twice a day (at 8:00 and 16:00 h). The diet contained citrus pulp, soybean hulls, urea, soybean or cottonseed meal and corn or sorghum silage as roughage source, and eventually sugarcane bagasse (Table 3).

These diets contained about 50–60% concentrate (on DM basis). The chemical composition of samples from the different farms is presented in Table 4. Sieving of diets and faeces, and fecal pH measurements were performed as described for the traditional feedlot system.

### 2.3. Clinical Examinations, Blood and Ruminal pH

During the visits, clinical examinations were carried out by veterinarians responsible for animal health. Briefly, twice a day, staff members trained in health care observed each pen carefully looking for animals showing any signs of health problems (e.g., isolation from rest of the group, atypical posture or gait, claudication, faecal score 1 or 2, bunk avoidance, etc.). In such cases, the staff team immobilized the animals and contacted a veterinarian to perform a clinical examination, following recommendations of Dirksen [12] for diseases of the digestive system. All information gathered during clinical examinations were used to make a diagnosis of the problem. 

At each visit to feedlots, jugular blood samples from 5–8 animals of the groups A1, A2 and F were collected, and blood lactate concentration was measured using reactive test strips for the enzymatic determination of lactate and a portable lactate analyser. In addition, in the feedlot for slaughter, samples of rumen fluid were collected from 5–8 animals from groups A1, A2 and F via ruminocentesis, approximately 1 h after feeding, to measure ruminal pH with a handheld pH meter. The animals raised in the sire production system had higher values, and their owners did not authorize rumen fluid collections by ruminocentesis after being warned about the small risk of peritonitis. 

### 2.4. Reference Data from Beef Cattle Raised Exclusively on Pasture

Samples of ruminal fluid and faeces (taken directly from the rectum or soon after natural defecation) were collected from 19 adult beef cattle raised exclusively in good quality *Urochloa decumbens* or *U. brizantha* pastures in seven different farms. These animals were subjected to a previous clinical examination and were diagnosed as healthy. The samples were evaluated to provide a reference value for ruminal (ruminocentesis) and fecal pH values, and for the particle size of faeces from healthy cattle. During these examinations, blood was also collected to estimate lactate concentration as previously described.

### 2.5. Data Analysis

For interpretation of data, averages were obtained along with maximum and minimum values (amplitude) found for each parameter. Such case studies should be conducted to investigate complex environments, held in natural contexts and applied to contemporary matters. This approach is ideal for applications in field-oriented studies [13]. It is important to emphasise that case studies do not have randomization nor experimental control; therefore, hypothesis tests were not conducted for any variables in this paper.

## 3. Results

The NDF levels varied greatly between the diets of animals targeted for slaughter after the feeding period in comparison to those fed to be sold as sires. In the feedlot for slaughter, the NDF levels were higher in the diets of the two adaptation phases than in the finishing phase, which had about 420 g/kg DM. In the sire production system, the NDF contents were among 475–531 g/kg DM. In the feedlot for slaughter, Mombaça silage had particles larger than 8 mm in size between 20.1 and 22.3% (Table 5 and Table 6). In contrast, the finishing diets had between 8.9 and 11.1% of particles larger than 8 mm, in both evaluation periods (Table 5 and Table 6). In feedlots that produced sires, the diets always had between 14.1 and 19.8% of the ingested particles larger than 8 mm (Table 7).

The percentages of particles in the Mombaça silage and in other diets greater than 8 mm and smaller than 4 mm are presented in Figure 1.

The pH values of rumen fluid and faeces were lower in animals from the feedlot for slaughter compared to animals raised on pasture or those fed to be sold as sires (Table 8). On the other hand, blood lactate was higher in cattle confined for slaughter.

Table 9 and Table 10 and Figure 2 show data on cattle fecal particle size. The feedlot diets for slaughter always generated a greater fecal escape of particles larger than 4 mm in relation to the two sire production systems.

## 4. Discussion

The concentration of NDF in feedlot diets for slaughter was slightly lower (Table 2) than the average value reported by Detmann et al. [15] for feedlot cattle in Brazil (484.7 ± 124). In the two feedlots systems for production of sires, the [NDF] ranged between 475–531 g/kg DM, due to the use of a forage:concentrate ratio close to 50%, a range considered safe to minimize the risks of digestive disorders [16]. Indeed, there were no detected cases of ruminal acidosis or bloat in any animal in the feedlots for sire production. A study incorporating results from 14 data sets from cows grazing exclusive pastures indicated that 41.1% NDF (on a DM basis) was required to maintain rumen pH at 5.8 to 6.0 [17]. De Campaneere et al. [18] demonstrated that NDF concentrations of 30% to 40% are necessary when it is used extremely rapidly fermented diets, such as those based on corn silage.

Although the finishing diets had 417–429 g NDF per kg of DM of which 336–347 g was peNDF, a value considered within the adequate range for zebu beef cattle (200–300 g/kg DM), many cattle fed this diet had ruminal acidosis, laminitis and bloat as observed in diagnoses during the field clinical examinations and confirmed by ruminal and fecal pH measurements [1]. The latter findings indicate that [NDF] and peNDF levels alone cannot be used to prevent SARA in high-grain diets for zebu cattle. This idea is reinforced by the fact that chewing activity can increase with increasing intake of peNDF without elevating ruminal pH, particularly when diets contain highly fermentable carbohydrate sources [19,20,21] because the concept of peNDF does not account for differences in ruminal fermentability of feeds, which can have a major effect on ruminal pH [22,23]. These many cases of ruminal acidosis were probably due to the high inclusion of steam-flaked corn/sorghum and the absence of buffers and ionophores in this finishing diet. Despite this, during the field inspections, the authors found cattle suffering from heat stress and in many pens feed competition in the in bunks at the time of feeding was detected. Heat stress causes a considerable change in the daily intake pattern, with greater preference for the ingestion of grains to the detriment of forage [3], and also reduces dry matter intake and generates great variation in feed intake and competition for feed in the bunks across days. These factors, perhaps, also affected the occurrence of SARA in the feedlot for slaughter. 

Ruminants can sort through feed, as demonstrated by DeVries et al. [24] in lactating dairy cows which had higher degrees of sorting against longer forage particles and for smaller grain concentrate particles when fed a low forage diet. The stud animals had an intake of 13 up to 17 g of NDF per kg of LW, which are values close to the averages for feedlot cattle found by Detmann et al. [15] using data from 181 experimental treatments published in 45 papers. 

In the feedlot of cattle for slaughter, the amount of forage ingested was reduced weekly, being A1 > A2 > F (Table 1). An important aspect was that the inclusion of steam-flaked corn increased from A1 to the finishing diet. The use of steam-flaked corn is aimed at increasing ruminal starch digestion and reducing its loss in faeces [25]. However, this greatly increases the risk of ruminal acidosis; a fact frequently observed in clinical examinations of the animals in finishing [1]. Therefore, diets with steam-flaked corn should be added with buffers and ionophores, which was not reported/found during the phase of data collection (Table 1). It is worth remembering that limestone, added in these two feedlot diets, has very little buffering action in the rumen [26].

Based on clinical examination, ruminal [27] and fecal pH values [1], as well as serum lactate concentration, which clearly indicated clinical and subacute rumen acidosis in animals confined for slaughter, the total diets in these feedlots should have been formulated to guarantee more than 15% of particles larger than 8 mm or 30–40% of particles close to 4mm length, based on the feedlot systems for sire production and reference data from cattle raised in pastures. This 15% empirical value is also based on the fact that animals confined for sale as sires did not have digestive disorders, and their total diets always had more than 14% of particles larger than 8 mm (Table 7 and Figure 1). This measure, associated with the use of other fibrous concentrates (e.g., whole cottonseed, citrus pulp and soybean hulls) and an adequate forage:concentrate ratio, could greatly minimize digestive problems in cattle confined for slaughter.

Subacute or chronic ruminal acidosis is best described as a syndrome related to a fermentative disorder of the rumen due to high intake of highly digestible starch (and other non-fibrous carbohydrates) and lower intake of NDF and physically effective NDF [28]. It involves a lowering of ruminal pH below pH 5.6 [27,29], which was found in cattle fed A2 and F diets in the feedlot for slaughter (Table 8). Based on clinical examination of many cattle with subacute rumen acidosis, the authors believe a fecal pH lower than 6.30 indicates that cattle were undergoing some degree of subacute rumen acidosis. This threshold value is close to that found by Osbourne et al. [30]. Thus, the fecal pH of the animals fed A2 and F diets coincided with the fact that many cattle from these diets were diagnosed with ruminal acidosis during field clinical examinations.

The value of 9.13 mmol of blood lactate/L found in cattle in the traditional feedlot system was potentially dangerous, being a predisposing factor for systemic metabolic acidosis [31], and could be the cause of many sudden death cases. During field inspections and data collection in this study, it was observed that several animals were seen isolated from others, but with no changes or signs of any problems and, after a certain time, were found dead by staff members. Unfortunately, no cases where the animal was in extremis were found, so that blood was collected, and lactate could be measured. Maruta and Ortolani [32] induced acute ruminal acidosis in Jersey and Gir cattle and found 11.7 and 6.8 mmol/L of blood lactate, respectively, for the two breeds. The reference value of healthy cattle is 0.60 up to 2.2 mmol/L [33]. According Meléndez et al. [34] the values obtained with handheld lactate meters are not comparable to concentrations obtained with a bench type approach used in the laboratory. This may explain why healthy cattle raised on pastures in the current work had values above those considered normal (Table 8).

Regarding the particle size distribution of the diets in the two feedlots to produce sires, there was always a higher proportion of feed particles larger than 8 mm (Table 7, Figure 1). In contrast, the diets used in the feedlots for slaughter always had higher proportions of particles with less than 4 mm, irrespective of the year evaluated (Table 5 and Table 6). These differences, associated with the data obtained in the clinical examinations, is one of the reasons for the null cases of ruminal acidosis in the two sire production systems in relation to the numerous cases verified in the feedlot system for slaughter [1]. It is important to emphasise that the main objective when feeding future sires is to help them reach a target LW to mate with cows soon after their purchase and arrival on cow-calf operations. It would be illogical to feed these animals inappropriately and risk the development of rumen acidosis and its common sequalae laminitis (coriosis), particularly because this claw lesion negatively affects the locomotor capacity of the animals, making them unable to mate during the breeding season.

In both feedlot systems for sire production, the greater inclusion of pelleted citrus pulp (Table 2) ensured a more “fibrous” diet and higher ruminal pH because citrus pulp is rich in pectin that, when fermented in the rumen, does not generate lactic acid, which is the reducing agent of ruminal pH [35]. The results of Zebeli et al. [28] demonstrated that fine chopping of roughage to a theoretical particle size of 4 to 6 mm adversely affected rumination activity and rumen fermentation in dairy cows receiving relatively large amounts of concentrate (50 to 60% DM); values much lower than those routinely used in beef cattle feedlots in Brazil, such as the one of this study (80–85% DM). Thus, current recommendations for particle size distributions appear to be adequate for high-forage diets, low in rapidly fermentable starch, but these recommendations will not eliminate the risks of the SARA in feedlot cattle fed low forage:concentrate diet, as seen in this study.

For practical purposes, the data in Table 9 and Table 10 were interpreted as the percentage of fecal particles larger than 4 mm and smaller than 1.18 mm. The proportion of fecal particles larger than 4 mm in size increased as the cattle ingested more concentrate in their diet (Table 9 and Table 10 and Figure 2). This is due to the fact that concentrates increase digesta passage rate through the digestive tract, cause subacute ruminal acidosis (which, in turn, reduces the activity of the fiber-digesting microbiota) and reduce rumination activity (which decreases the particle fragmentation rate). That is, in the present study, more than 4% of fecal DM as particles larger than 4 mm was always associated with cases of SARA in cattle fed high amounts of concentrate feeds (Figure 2), since the stud animals in the feedlot for sire production, and cattle kept only on pastures, had no diagnosis of SARA and did not excrete more than 4% of fecal particles larger than 4mm (Figure 2). Nørgaard and Sehic [36] found that less than 5% of fecal particles were larger than 5 mm in cows fed grass silage with different theoretical chopping lengths. Large particle fiber in faeces is an indication of short retention time of feed in the rumen and poor reduction size of particles by rumination and microbial fermentation.

Perhaps due to the lack of standardization of the forms of particle size separation in the faeces, the literature reports contrasting values. Poppi et al. [7] mentioned that less than 5% of fecal particles from cattle and sheep raised on pastures were retained in sieves larger than 1.18 mm, and Maulfair et al. [37] found more than 36% of fecal particles larger than 1.18 mm. The value 1.18 mm was adopted as the critical size, below which the food particles would pass through the reticulo-omasal orifice, reach the abomasum and be excreted in faeces [4,7]. In the present study, the proportion of fecal particles larger than 1.18 mm was lower in cattle raised on pastures (14.3%) and higher in animals fed the finishing diet in the feedlot for slaughter (27.7%) (Table 10). Suarez-Mena et al. [38] found, in heifers fed high or low amount of forage, values between 6.5 and 13.8% of fecal DM retained in 1.18 mm sieves. Faichney [39] described a model in which the limit size of particles that leave the reticulum-rumen was postulated to be between 1–2 mm, while the model developed by Mertens et al. [40], postulated a size of 3–4 mm. Once larger particles escape to the abomasum, it is unlikely they will be reduced in size as they travel through the intestines until they are excreted in faeces. Researchers from the USA, Canada and Japan have studied particle size of diets and their impact on rumen metabolism, and have clearly shown that the critical threshold for feed particles escaping the rumen of high-producing cows is greater than 1.18 mm and more in the range of 4 mm [14]. A recently published work demonstrated that the variations in pH and particle size of faeces in dairy cattle were due to changes in dietary starch [41]. The latter study corroborates our findings and recommendations for the use of these ancillary tests as on-farm tools for assessing the risk of ruminal acidosis. As highlighted by Thomson et al. [42], worldwide research and development in agriculture can only be maximised through adequate extension methods to effectively reach the rural communities they serve. Case studies, such as in the current work, are often conducted in partnership with rural producers, and represent an ideal approach to reach out to those rural communities and demonstrate the use of existing knowledge applied into real and practical contexts.

Another important type of work is presented in the latest surveys of nutritional recommendations and management practices adopted by feedlot cattle nutritionists [43,44]. The latter studies indicated that diets, such as those used in the finishing feedlots in the current study, are in use in other commercial operations in Brazil. However, in both surveys there was no mention to the use of on-farm tools such as the ancillary tests proposed here. In summary, we believe that none of the tests that are currently available could be considered as an independent gold standard to diagnose SARA in large Brazilian feedlots. Ancillary tests, such as ruminal and faecal pH and particle size distribution in the faeces can generate good information, especially when used in combination with nutritional composition (e.g., %NDF and peNDF) and a series of best practice management protocols (i.e., minimize heat stress, adequate bunk space to avoid feed competition, correct adaptation to the to lower roughage diets, frequent cleaning of water troughs, and regular removal of mud and manure excess).

## 5. Conclusions

Based on the results obtained in the present study, under conditions of diets and management typically used in Brazilian feedlots, it is suggested that subacute rumen acidosis in feedlots may occur in situations where more than 4% of faecal dry matter is excreted as particles larger than 4 mm, associated with diets with less than 15% of particles smaller than 8 mm and faecal pH under 6.3. Given the difficulty in identifying cases of subacute rumen acidosis in large feedlots, these thresholds could be an important support aid the establishment correct diagnosis and to minimize economic and animal welfare losses by ruminal disorders linked to errors in nutritional management. Thus, formulating diets on the basis of %NDF and peNDF without accounting for the fermentability of the diet may not totally eliminate the risks of ruminal acidosis in the situations where diets are rich in rapidly fermented starch, as was the case seen in the feedlot for slaughter where the animals were fed with a low forage:concentrate ratio in the diet and the concentrate was mainly constituted by steam-flaked corn or sorghum.

## Figures and Tables

**Figure 1 animals-12-03114-f001:**
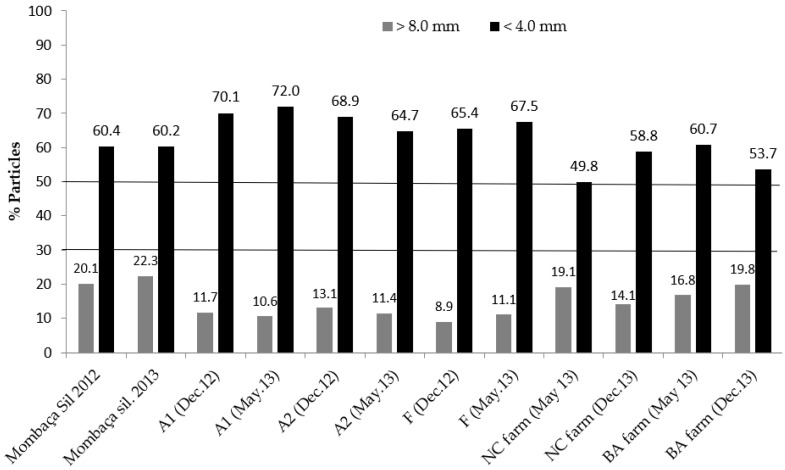
Percentages of particles larger than 8 mm and smaller than 4.0 mm in roughage and total diets. The horizontal lines represent the ranges of ideal empirical values in which the diets would have the appropriate proportion of particles greater than 8.0 mm as indicated by Heinrichs et al. [14].

**Figure 2 animals-12-03114-f002:**
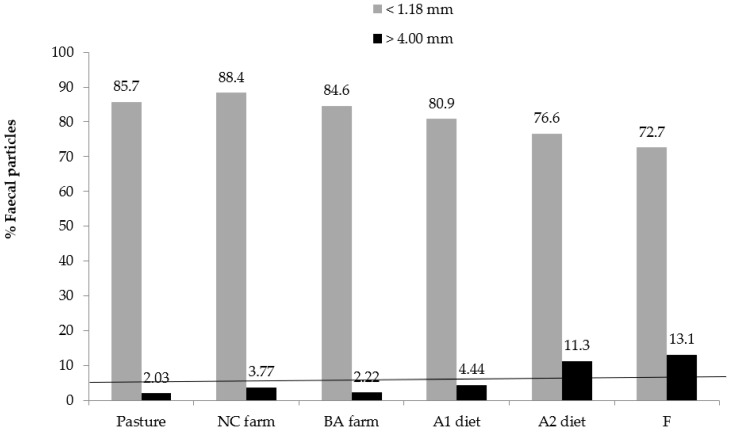
Relationship between the overall mean of percentage of fecal particles larger than 4 mm and smaller than 1.18 mm from cattle raised in feedlots and on pasture. The horizontal line determines the empirical value (4% of fecal dry matter as particles >4 mm), above which the animals had subacute ruminal acidosis, diagnosed based on clinical examinations performed in the field, associated with ruminal and fecal pH, fecal score and blood lactate values.

**Table 1 animals-12-03114-t001:** Ingredients of the diets of finishing feedlot (traditional system).

Feedstuffs	October 2012			May 2013		
(% dry matter)	A1	A2	F	A1	A2	F
Mombaça grass silage	52.0	39.8	17.5	22.6	15.7	3.88
Sugarcane bagasse	-	-	-	29.2	18.2	10.6
Steam-flaked corn	31.8	37.3	53.2	-	-	-
Soybean hulls	-	-	-	-	-	2.87
Soybean meal	4.10	8.10	10.2	5.90	5.00	-
Whole cottonseed	-	-	-	5.90	11.5	14.3
Steam-flaked sorghum	-	-	-	32.6	45.3	52.3
Sorghum meal	1.70	1.90	4.70	1.51	1.80	1.65
Soybean molasses	8.10	10.8	11.1	-	-	-
Optgen ^1^	-	-	-	0.20	0.20	0.25
Urea	0.60	0.51	1.10	0.50	0.61	1.00
KCl	-	-	-	0.20	0.20	0.37
Biopro ^2^	0.60	0.70	0.90	0.50	0.60	0.52
Limestone	0.60	0.50	0.90	0.50	0.60	1.00
NaCl	0.60	0.50	0.40	0.30	0.30	0.25
Water	-	-	-	-	-	11.0

^1^ Optgen = controlled release urea; ^2^ Biopro = probiotic + prebiotic; A1 = Adaptation diet 1; A2 = Adaptation diet 2; F = Finishing diet.

**Table 2 animals-12-03114-t002:** Chemical composition of the diets used in the traditional feedlot system (i.e., A_1_ = Adaptation diet 1; A_2_ = Adaptation diet 2; F = Finishing diet).

Item	Diet (October 2012)			Diet (May 2013)		
	A1	A2	F	A1	A2	F
DM ^1^ (g/kg as fed)	438	491	562	458	481	523
CP ^2^ (g/kgDM)	111	124	139	129	138	132
EE ^3^ (g/kgDM)	21.5	28.1	31.0	30.7	31.2	29.4
Ash ^4^ (g/kgDM)	62.1	73.5	46.4	58.3	63.1	55.7
NDF ^5^ (g/kgDM)	503	488	417	517	506	429
peNDF ^6^ (g/kg DM)	439	405	336	430	389	347
NFC ^7^ (g/kgDM)	304	286	367	281	293	351
P ^8^ (g/kgDM)	2.7	3.7	3.6	3.1	3.4	3.9
Ca ^9^ (g/kgDM)	6.2	6.9	5.7	6.6	6.1	6.2

^1^ DM = dry matter; ^2^ CP = crude protein; ^3^ EE = ether extract; ^4^ MM = mineral matter; ^5^ NDF = neutral detergent fiber; ^6^ peNDF = physically effective neutral detergent fiber; ^7^ NFC = non-fibrous carbohydrates; ^8^ P = phosphorus; ^9^ Ca = calcium.

**Table 3 animals-12-03114-t003:** Diets of the two feedlots for sire production. Values for a 500 kg animal.

	NC Farm (May/2013)		NC Farm (July/2013)		BA Farm (April/2013)		BA Farm (July/2013)	
Feed ingredients	DMI		DMI		DMI		DMI	
	(kg/d) (%)		(kg/d) (%)		(kg/d) (%)		(kg/d) (%)	
Sugarcane bagasse	5.00	47.2	4.00	38.4	-	-	-	-
Sorghum silage	-	-	-	-	5.76	50.0	5.30	47.1
Citrus pulp	3.52	33.2	4.10	39.3	4.40	38.2	4.50	40.0
Cottonseed meal	1.34	12.6	1.40	13.4	-	-	-	-
Proteic concentrate mix	-	-	-	-	1.34	11.5	1.40	12.5
Corn meal	0.62	5.88	0.80	7.70	-	-	-	-
Urea	0.12	1.17	0.12	1.15	0.04	0.35	0.04	0.35
Total	10.60		10.42		11.54		11.24	
DMI ^1^ (% liveweight)	2.1		2.1		2.3		2.2	

^1^ DMI = dry matter intake.

**Table 4 animals-12-03114-t004:** Average composition of the diets used in the two feedlots for sire production.

Items	NC Farm (May 2013)	NC Farm (July 2013)	BA Farm (April 2013)	BA Farm (July 2013)
DM ^1^ (g/kg as fed)	494	396	502	527
CP ^2^ (g/kgDM)	91.9	82.7	93.4	103.0
EE ^3^ (g/kgDM)	24.7	30.1	32.3	34.5
ASH ^4^ (g/kgDM)	25.6	31.2	29.8	30.1
NFC ^5^ (g/kgDM)	246	220	69.5	130
NDF ^6^ (g/kgDM)	512	531	475	502
peNDF ^7^ (g/kgMS)	446	439	421	412
P ^8^ (g/kgDM)	1.4	1.3	1.3	1.6
Ca ^9^ (g/kgDM)	3.7	3.3	2.8	3.5

^1^ DM = dry matter; ^2^ CP = crude protein; ^3^ EE = ether extract; ^4^ MM = mineral matter; ^5^ NFC = non-fibrous carbohydrates; ^6^ NDF = neutral detergent fiber; ^7^ peNDF = physically effective neutral detergent fiber; ^8^ P = phosphorus; ^9^ Ca = calcium.

**Table 5 animals-12-03114-t005:** Average of particle size distribution of the Mombaça silage and the diets of the traditional feedlot (October 2012). Values expressed in %.

Particles (mm)	Mombaça Silage		Diets	
		A_1_	A_2_	F
>8.0	20.1 (18.1–21.7)	11.7 (10.7–12.1)	13.0 (11.1–15.4)	8.9 (5.4–10.1)
4.0–8.0	19.4 (16.1–22.9)	18.4 (15.8–19.8)	18.1 (16.4–20.2)	25.9 (21.8–27.8)
2.0–4.0	46.1 (43.3–48.4)	53.3 (50.2–56.1)	53.9 (51.0–55.8)	48.9 (43.6–52.0)
<2	14.3 (11.2–16.9)	16.8 (14.6–17.9)	15.0 (12.7–17.2)	16.5 (12.8–18.4)
Total (%)	99.9	100.2	100.1	100.2

A_1_ = Adaptation diet 1; A_2_ = Adaptation diet 2; F = Finishing diet. Values in parentheses refer to the amplitude.

**Table 6 animals-12-03114-t006:** Average of particle size distribution of the Mombaça silage and the diets of the traditional feedlot (May 2013). Values expressed in %.

Particles (mm)	Mombaça Silage		Diets	
		A_1_	A_2_	F
>8.0	22.3 (19.0–24.5)	10.6 (8.7–12.9)	11.4 (10.1–12.8)	11.1 (9.1–12.5)
4.0–8.0	17.6 (15.9–18.7)	17.6 (12.8–19.1)	23.7 (21.6–25.8)	21.3 (20.3–23.2)
2.0–4.0	49.3 (42.7–51.1)	56.0 (53.4–59.1)	49.9 (45.9–52.2)	43.6 (41.0–45.6)
<2	10.9 (7.8–12.3)	16.0 (13.6–17.8)	14.8 (12.8–15.7)	23.9 (20.9–26.1)
Total	100.1	100.2	99.8	99.9

A_1_ = Adaptation diet 1; A_2_ = Adaptation diet 2; F = Finishing diet. Values in parentheses refer to the amplitude.

**Table 7 animals-12-03114-t007:** Average of particle size distribution of the diets of the feedlots for sire production. Values expressed in %.

Particles (mm)	NC Farm May 2013	NC Farm July 2013	BA Farm April 2013	BA Farm July 2013
>8.0	19.1 (17.1–21.0)	14.1 (12.9–15.8)	16.8 (13.7–18.2)	19.8 (18.1–21.8)
4.0–8.0	31.2 (29.2–32.8)	22.3 (21.1–24.0)	22.3 (21.1–24.8)	26.7 (25.1–28.6)
2.0–4.0	22.7 (20.9–23.8)	31.3 (31.2–34.7)	30.1 (28.5–32.1)	24.8 (22.8–26.1)
<2	27.1 (25.0–30.3)	32.2 (29.9–34.3)	30.6 (28.8–31.9)	28.9 (25.9–30.2)
Total (%)	100.1	99.9	99.8	100.2

Values in parentheses refer to the amplitude.

**Table 8 animals-12-03114-t008:** Overall averages of ruminal and fecal pH and blood lactate evaluated in cattle raised in the two feedlot systems and those raised in pastures without concentrate intake.

	Ruminal pH	Fecal pH	Blood Lactate (mmol/L)
A_1_ diet (*n* = 13)	5.68 (5.51–6.08)	6.41 (6.13–6.69)	6.23 (4.01–7.21)
A_2_ diet (*n* = 14)	5.39 (5.29–5.58)	6.33 (6.04–6.53)	6.10 (3.03–8.10)
F diet (*n* = 16)	5.34 (5.12–5.61)	6.26 (6.10–6.39)	9.13 (6.13–11.3)
Pasture (*n* = 19)	6.72 (6.61–6.93)	7.53 (7.12–7.81)	3.55 (2.17–4.93)
NC farm diet (*n* = 7)	nm	6.64 (6.33–6.74)	5.02 (3.19–6.67)
BA farm diet (*n* = 6)	nm	6.48 (6.29–6.67)	nm

A_1_ = Adaptation diet 1; A_2_ = Adaptation diet 2; F = Finishing diet; nm = not measured; Values in parentheses refer to the amplitude.

**Table 9 animals-12-03114-t009:** Overall mean of faecal particle size of the animals fed finishing diets for slaughter. Values expressed in %.

Particles (mm)	A_1_ Diet (*n* = 10)	A_2_ Diet (*n* = 10)	F Diet (*n* = 10)
>4.0	4.44 (3.21–5.02)	12.72 (10.2–15.1)	13.2 (10.9–15.4)
2.0–4.0	5.95 (4.02–6.29)	5.23 (4.38–6.23)	7.01 (6.11–8.23)
1.18–2.0	8.26 (7.21–9.74)	5.56 (4.12–7.05)	7.51 (6.34–8.56)
<1.18	81.3 (76.6–84.1)	76.5 (73.3–79.1)	72.3 (70.7–74.1)
Total	100.0	100.0	100.0

A_1_ = Adaptation diet 1; A_2_ = Adaptation diet 2; F = Finishing diet. Values in parentheses refer to the amplitude.

**Table 10 animals-12-03114-t010:** Overall means of fecal particle size of cattle submitted to the diets of the two sire production systems and of the animals raised on pasture only. Values expressed in %.

Particles (mm)	NC Farm (*n* = 10)	BA Farm (*n* = 10)	Cattle Raised on Pastures (*n* = 10)
>4.0	3.79 (2.91–5.03)	2.22 (1.59–2.87)	2.03 (1.70–4.67)
2.0–4.0	2.74 (1.87–4.31)	6.61 (5.12–7.61)	5.07 (3.36–8.01)
1.18–2.0	4.84 (3.86–5.93)	6.54 (3.38–7.97)	7.19 (5.24–9.87)
<1.18	88.6 (84.1–93.4)	84.6 (81.2–90.1)	85.7 (79.3–91.0)
Total	100.0	100.0	100.0

Values in parentheses refer to the amplitude.

## Data Availability

All data supporting reported results can be shared upon request directly to our corresponding author.

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
