# Peer review of "Physicochemical Evaluations of Diets, Rumen Fluid, Blood and Faeces of Beef Cattle under Two Different Feedlot Systems"

_animals, 2022, doi:10.3390/ani12223114_

Round 1

Reviewer 1 Report

See the attachment

Author Response

Comments from Reviewer 1:

 General comments:

The study reported the "Physicochemical evaluations of diets, rumen fluid, blood lactate and faeces of beef cattle under two different feedlot systems in Brazil". To assessed it, authors used in two distinct feedlot systems, to understand the causes and correlations to digestive disorders in those production systems. However, protocols of feed management and slaughtered are not adequate methodologies and mainly discussed their results are not enough and clear. I feel that your work is preliminary and does not provide enough new scientific findings in Animals wishes to have.

Response:

*** Thank you for your concern. We emphasise that the proposed methodology is novel, and we strongly believe the current work presents solid evidence (and references) to be awarded publication. A very recent example of a work of the same nature has been published by Korrhami et al. in the Research in Veterinary Science Journal https://doi.org/10.1016/j.rvsc.2022.10.001  In order to clarify our intent, the latter work has been added as a reference in ***Lines 394-398 that now read: “A recently published work demonstrated that the variations in pH and particle size of faeces in dairy cattle were due to changes in dietary starch [41]. The latter study corroborates with our findings and recommendations for the use of these ancillary tests as on-farm tools for assessing the risk of ruminal acidosis.”

Specific comments:

I have some questions that I don't understand, so please answer it for me.

In article, there are many references cited more than 10 years.

Response:

*** We have indeed included some of the classic references of our area and believe these should be there. Perhaps if the reviewer specifies which references, then we can reassess if any could be replaced. However, as explained above, the older references included are considered classic in this field of research.

Data were collected in October 2012 and May 2013; the data is not up to date ... ???. Materials and Methods:

Response:

*** We appreciate your concern. The first paper from a series of work that occurred parallel to this study was published in 2016 and was cited in the current manuscript as reference [1] “Malafaia, P., Granato, T.A.L., Costa, R.M., Souza, V.C., Costa, D.F.A., and Torkania, C.H. Major health problems and their economic impact on beef cattle under two different feedlot systems in Brazil. Pesquisa Veterinaria Brasileira 2016, 36, 837-843.” Since its publication, the paper has been cited twenty times. The data used in the current manuscript was used with a focus on a specific niche of professionals and has the potential to change the existing practices to assess digestive disorders on farm. Therefore, we strongly believe the data is not only up to date, but also very relevant. If you have a chance to read the latest surveys of nutritional recommendations and management practices adopted by feedlot cattle nutritionists in Brazil from 2019 and 2021 [https://doi.org/10.1139/cjas-2018-0031 and https://doi.org/10.37496/rbz5020200189] you will notice that the diets used in the finishing feedlot in our work clearly reflect diets currently used in many other commercial operations. As previously emphasised, the authors of the current work strongly believe these findings have the potential to strongly improve practices currently in place in beef cattle feedlot operations worldwide.

To further emphasise the importance of the current study, the following has been added to ***Lines 404-408: “Another important type of work is presented in the latest surveys of nutritional recommendations and management practices adopted by feedlot cattle nutritionists [43,44]. The latter studies indicated that diets like those used in the finishing feedlots in the current study are in use in other commercial operations in Brazil. However, in both surveys there is no mention to the use of on-farm tools such as the ancillary tests proposed here.”

Experimental design

Line 195-201; Data analysis is not clear "what is design". The main change I would recommend to the authors is a more detailed description of the management feeding system etc. pasture group as grazing, it is supplemented with concentrate or not ??? ..... ..

Response:

***As indicated in ***Lines 196-197: “For interpretation of data, the averages were described along with maximum and minimum values (amplitude) found for each parameter”. It is important to note that the current work represents a combination of case studies; hence why the statistical design does not conform to more traditional ways used by research centres/institutions not working in extension. To emphasise once again the importance of works of this nature, the following paragraph has been added to ***Lines 398-403: “As highlighted by Thomson et al [43], worldwide research and development in agriculture can only be maximised through adequate extension methods to effectively reach the rural communities they serve. Case studies such as in the current work are often conducted in partnership with rural producers, what represents an ideal approach to reach out to those rural communities and demonstrate the use of existing knowledge applied into real and practical contexts.”

***As for the cattle grazing, there was no use of supplement. ***Line 187 reads: “2.4. Reference data from beef cattle raised exclusively on pasture”. In addition, in ***Lines 189-190 the following paragraph can be found: “cattle raised exclusively in good quality Urochloa decumbens or U. brizantha pastures”.

Line 126-127; Why the authors did not analyzed fecal sample immediately after collected sample. ???? After thawing was analyzed pH maybe changed ....... .it is very important to explain.

Response:

***The analyses were performed at the university, approximately 1400 km from where the field work was conducted. Hence why the samples had to be frozen to avoid issues such as pH changes due to continued fermentation. Our method was tested in house with a large number of samples being analysed on site and subsequently frozen for further analysis. Our results proved the method was fine.

Table 9, the authors should be a more detailed description of the A2 diet and F diet are similar but A2 greater fecal escape of particles larger than 4 mm than Al ??.

Response:

***The adaptation diets are different and therefore the escape of particle sizes should be expected to be different.

In conclusion: The authors are mainly focused on physicochemical characteristics of diets and faeces, along with data of rumen fluid and blood lactate collected in two distinct feedlot systems. These parameters did not enough to conclusion, and I think it is not appropriate for ANIMALS publication. I think it is very short article, please provide more information such as feed intake and other important parameter. The final decision, of course, rests with the Editors of Animals Journal.

***As discussed above, the proposed methodology is novel. We respect the reviewer’s opinion but in contrast, we strongly believe the current work presents solid evidence (and references) to be awarded publication. As for intake, commercial operations rarely measure individual intake. For the objectives of the current work, information of group intake would not add much.

Reviewer 2 Report

The researchers evaluated the physicochemical characteristics of the diet and feces collected in two different feeding systems in Brazil, as well as rumen fluid and blood lactate data, to understand the causes and correlations of digestive disorders in these production systems. The comments are shown below:

1. The authors reported the data of the samples collected in 2012 and 2013. It has been about 10 years since then. Has the feeding mode of Brazilian beef cattle changed? Can the conclusions were drawn by the author guide beef cattle breeding today? Ask the author to introduce in detail in the background.

2. How many groups are 10 fecal samples collected? Is it representative? The author needs to describe the screening of fecal samples in detail.

3. How do rumen juice and blood samples correspond to fecal samples to judge the state of animals?

4. An important problem in this paper is that the sample size is too small to represent the Brazilian beef cattle population.

Author Response

Comments from Reviewer 2:

The researchers evaluated the physicochemical characteristics of the diet and feces collected in two different feeding systems in Brazil, as well as rumen fluid and blood lactate data, to understand the causes and correlations of digestive disorders in these production systems. The comments are shown below:

***Thank you very much. The authors really appreciate the effort and time you spent on this.

Specific comments:

  1. The authors reported the data of the samples collected in 2012 and 2013. It has been about 10 years since then. Has the feeding mode of Brazilian beef cattle changed? Can the conclusions were drawn by the author guide beef cattle breeding today? Ask the author to introduce in detail in the background.

***That’s a good point. The nutritional recommendations and practices in Brazil have been changing as indicated by a series of surveys of feedlot nutritionists. However, the diets used in the current work are still in line with finishing diets of recent years. More importantly, the current work presents two contrasting scenarios, one with a higher NDF concentration and one with a much lower NDF concentration. Brazil’s feedlot industry is moving at increasing rates towards the latter scenario, hence why the findings in the current work are so crucial. To clarify this point, the following paragraph has been added to ***Lines 404-408: “Another important type of work is presented in the latest surveys of nutritional recommendations and management practices adopted by feedlot cattle nutritionists [43,44]. The latter studies indicated that diets like those used in the finishing feedlots in the current study are in use in other commercial operations in Brazil. However, in both surveys there is no mention to the use of on-farm tools such as the ancillary tests proposed here.”

  1. How many groups are 10 fecal samples collected? Is it representative? The author needs to describe the screening of fecal samples in detail.

Thank you for noticing. We had not included this information. This has been corrected. Now ***Lines 123-125 read: “The latter samples were checked to assure they had a similar faecal score and were collected from cattle with similar LW that had defecated at the moment of collections.”

  1. How do rumen juice and blood samples correspond to fecal samples to judge the state of animals?

*** The animal groups in A1, A2 and F had similar LW and were receiving the same diets, hence why we anticipated that any animal within each of those groups would have similar rumen pH and blood lactate values. Therefore, linking that data to faecal characteristics seemed appropriate. To add more to this discussion, we included a recent reference in ***Lines 394-398 that now read: “A recently published work demonstrated that the variations in pH and particle size of faeces in dairy cattle were due to changes in dietary starch [41]. The latter study corroborates with our findings and recommendations for the use of these ancillary tests as on-farm tools for assessing the risk of ruminal acidosis.”

  1. An important problem in this paper is that the sample size is too small to represent the Brazilian beef cattle population.

To avoid confusion, we have removed the word “Brazil” from the title. Note that the objectives of the current work were to demonstrate the usefulness of these ancillary tests in two distinct feedlot systems.

Reviewer 3 Report

The purpose of the reviewed work was to the fecal particle size distribution, associated with ruminal and faecal pH, can be an indicator of the rumen health of feedlot cattle in the finishing phase. Data were collected in October 2012 and May 2013 from a feedlot located in the state of Goiás, Brazil, that bought in and finished about 80,000 head per year. The subject of article is adequate to its content. The article is written in a clear and understandable way. The authors have contributed a lot of work, but have not avoided some shortcomings:

- signatures to Figure1 should be standardized (with or without a full stop)

- lines 234-235 it is enough to refer to Table 8 once

-line 259 - please expand the abbreviation in brackets or remove the repetition

- [NDF] - I don't think it should be used in the text in parentheses

The comments do not diminish the value of the paper, and I think that with the recommended corrections, the manuscript may be ready for publication.

Author Response

Comments from Reviewer 3:

The purpose of the reviewed work was to the fecal particle size distribution, associated with ruminal and faecal pH, can be an indicator of the rumen health of feedlot cattle in the finishing phase. Data were collected in October 2012 and May 2013 from a feedlot located in the state of Goiás, Brazil, that bought in and finished about 80,000 head per year. The subject of article is adequate to its content. The article is written in a clear and understandable way. The authors have contributed a lot of work, but have not avoided some shortcomings:

***Thank you. We appreciate your input and below have tried to address your considerations accordingly.

Specific comments:

- signatures to Figure1 should be standardized (with or without a full stop).

*** Thank you. Modifications done as suggested. All Tables and Figures are now with full stop.

- lines 234-235 it is enough to refer to Table 8 once

***Thank you. We’ve removed the second reference to Table 8.

-line 259 - please expand the abbreviation in brackets or remove the repetition.

***Thank you. The repetition has been removed as suggested.

- [NDF] - I don't think it should be used in the text in parentheses.

***The parentheses have been removed.

Reviewer 4 Report

1. In the abstract part, the language needs to be refined and organized again, and specific parameter indicators and values need to be listed.

2. The faeces should be collected from the rectum. Collecting faeces on the ground will delay the collection time and affect the pH of faeces.

3. The PH of feces is closely related to the composition. Is the composition of feces determined?

4. Are the blood biochemical indicators tested? The correlation between SARA and rumen, fecal pH, fecal particle size should be analyzed in combination with blood biochemical indicators and fecal composition.

Author Response

Comments from Reviewer 4:

General comments:

  1. In the abstract part, the language needs to be refined and organized again, and specific parameter indicators and values need to be listed.

***Thank you. The Abstract has been refined and reorganised. Now ***Lines 19-32 read: “The physicochemical characteristics of diets and faeces was evaluated in combination with data of rumen fluid and blood lactate collected from two distinct feedlot systems in Brazil to understand the causes and correlations to digestive disorders in those production systems. The data was collected during two visits to a finishing system which fed about 80,000 head per year, and four visits to two properties that fed 150 to 180 straight bred Nellore bulls per year to be sold as stud cattle. The findings suggest that ruminal acidosis occurred where there was high intake of starch-rich concentrate, and that subacute rumen acidosis (SARA) was most likely occurring in situations where more than 4% of faecal dry matter was excreted as particles larger than 4 mm. The latter were associated to diets with less than 15% of particles smaller than 8 mm and faecal pH under 6.30. It’s concluded that ancillary tests, such as ruminal and faecal pH and particles size distribution in the faeces can potentially be used in combination with information on diet nutritional composition and a series of best practice management protocols to increase not only animal productivity, but also reduce the risks of SARA and ensure the welfare of animals.”

  1. The faeces should be collected from the rectum. Collecting faeces on the ground will delay the collection time and affect the pH of faeces.

***That is a very important consideration. However, not so simple to be done in large commercial operations. To further clarify what protocols were in place to avoid such issue, the following paragraph has been added in ***Lines 123-125: “The latter samples were checked to assure they had a similar faecal score and were collected from cattle with similar LW that had defecated at the moment of collections.”

  1. The PH of feces is closely related to the composition. Is the composition of feces determined?

***Thank you. You are correct; however, our hypothesis was that ancillary tests would generate fast results correlated to ruminal acidosis and therefore the chemical composition of faeces was left aside in the current work.

  1. Are the blood biochemical indicators tested? The correlation between SARA and rumen, fecal pH, fecal particle size should be analyzed in combination with blood biochemical indicators and fecal composition

***Blood lactate was used as an indicator in the current work and proved to be very useful.

Round 2

Reviewer 1 Report

None

Reviewer 2 Report

I have no more comments.